# SARS-CoV-2 escape from cytotoxic T cells during long-term COVID-19

Oksana V. Stanevich[1,10], Evgeniia I. Alekseeva [2,10] ✉, Maria Sergeeva [1], Artem V. Fadeev [1], Kseniya S. Komissarova [1], Anna A. Ivanova [1], Tamara S. Simakova [3], Kirill A. Vasilyev [1], Anna-Polina Shurygina[1], Marina A. Stukova[1], Ksenia R. Safina [2], Elena R. Nabieva [4], Sofya K. Garushyants [4,9], Galya V. Klink[4], Evgeny A. Bakin[5,6], Jullia V. Zabutova[7], Anastasia N. Kholodnaia [5,7], Olga V. Lukina [5], Irina A. Skorokhod[7], Viktoria V. Ryabchikova [7], Nadezhda V. Medvedeva[7], Dmitry A. Lioznov[1,5], Daria M. Danilenko [1], Dmitriy M. Chudakov [2,8], Andrey B. Komissarov [1] & Georgii A. Bazykin [2,4] ✉

Evolution of SARS-CoV-2 in immunocompromised hosts may result in novel variants with changed properties. While escape from humoral immunity certainly contributes to intra-host evolution, escape from cellular immunity is poorly understood. Here, we report a case of long-term COVID-19 in an immunocompromised patient with non-Hodgkin's lymphoma who received treatment with rituximab and lacked neutralizing antibodies. Over the 318 days of the disease, the SARS-CoV-2 genome gained a total of 40 changes, 34 of which were present by the end of the study period. Among the acquired mutations, 12 reduced or prevented the binding of known immunogenic SARS-CoV-2 HLA class I antigens. By experimentally assessing the effect of a subset of the escape mutations, we show that they resulted in a loss of as much as ~1% of effector CD8 T cell response. Our results indicate that CD8 T cell escape represents a major underappreciated contributor to SARS-CoV-2 evolution in humans.

SARS-CoV-2 evolution in the global human population has involved the accumulation of mutations that increase viral transmissibility and cause immune escape[1]. A similar set of mutations is also observed in intra-host evolution of SARS-CoV-2 during long-term COVID-19, particularly in immunocompromised patients[2–13] treated with monoclonal antibodies and/or convalescent plasma[10,14]. Both the mutations that spread rapidly in the general population and those that accumulate in intra-host evolution facilitate entry of viral particles into host cells[15,16] and/or affect binding sites of neutralizing antibodies[2–4,17,18].

In addition to escape from humoral immunity, SARS-CoV-2 evolution can also involve escape from cellular response. Indeed, the landscape of antigen presentation, which is determined by the particular set of patient's HLA alleles, has a significant impact on the course of COVID-19[19–21]. Consistently, in the general population, SARS-CoV-2 acquires changes that reduce binding of viral antigens to HLA I molecules, weakening antigen recognition by corresponding cytotoxic T cell clones[22,23]. However, at the population level, accumulation of T cell escape mutations is complicated by the diversity of HLA molecules, and

[1]Smorodintsev Research Institute of Influenza, Saint-Petersburg, Russia. [2]Skolkovo Institute of Science and Technology (Skoltech), Moscow, Russia. [3]Parseq Lab Co. Ltd, Saint Petersburg, Russia. [4]A.A. Kharkevich Institute for Information Transmission Problems of the Russian Academy of Sciences, Moscow, Russia. [5]First Pavlov State Medical University, Saint-Petersburg, Russia. [6]Bioinformatics Institute, Saint Petersburg, Russia. [7]City Hospital 31, Saint-Petersburg, Russia. [8]Center for Precision Genome Editing and Genetic Technologies for Biomedicine, Institute of Translational Medicine, Pirogov Russian National Research Medical University, Moscow, Russia. [9]Present address: National Center for Biotechnology Information, National Library of Medicine, National Institutes of Health, Bethesda, MD, USA. [10]These authors contributed equally: Oksana V. Stanevich, Evgeniia I. Alekseeva. ✉e-mail: evg.alekseeva93@gmail.com; g.bazykin@skoltech.ru

the changes that lie at the origin of SARS-CoV-2 variants of concern are insignificant for CD4 and CD8 T cell reactivity in most patients[24]. Long-term COVID-19 may facilitate T cell escape, as the selection pressure favoring such an escape remains constant throughout the disease. Indeed, intra-host escape from T cells of both types was described for other long-term infections including HIV-1 and hepatitis C[25–29].The

Here, we show a case of long-term evolution of SARS-CoV-2 in an immunocompromised patient involving escape from T cell-mediated immunity.

## Results

### Case description

Patient S (Supplementary Note 1), a female previously diagnosed with Non-Hodgkin's diffuse B-cell lymphoma IV stage B, tested positive for SARS-CoV-2 for the first time on April 17, 2020. In the preceding week, she had had close contact with patient A, who later died of COVID-19 pneumonia; paraffin blocks with post-mortem material of patient A

were subsequently analyzed for SARS-CoV-2 by PCR, followed by RNA extraction and sequencing, as a probable source of infection. Patient S has undergone three periods of positive tests, alternating with two periods of negative tests, between April 17, 2020 and March 1, 2021, spanning a total of 318 days (see Fig. 1a, Supplementary Table 1). She had symptoms of severe COVID-19 between June 6 and September 1, 2020 (Supplementary Fig. 1a, b), and again between January 9 and March 1, 2021 (Supplementary Fig. 1c), including subfebrile fever and pneumonia with typical COVID-19 patterns. We isolated live viruses from swab samples obtained in both of these periods (August 20, 2020 and February 19, 2021).

Between April 30, 2020 and February 16, 2021, patient S received several cycles of chemotherapy under several different regimens, including monoclonal antibody rituximab. Courses of chemotherapy were typically followed by a decrease in white blood cell counts, especially lymphocyte and neutrophil counts, to values below the normal range (Supplementary Fig. 2). On December 28, 2020,

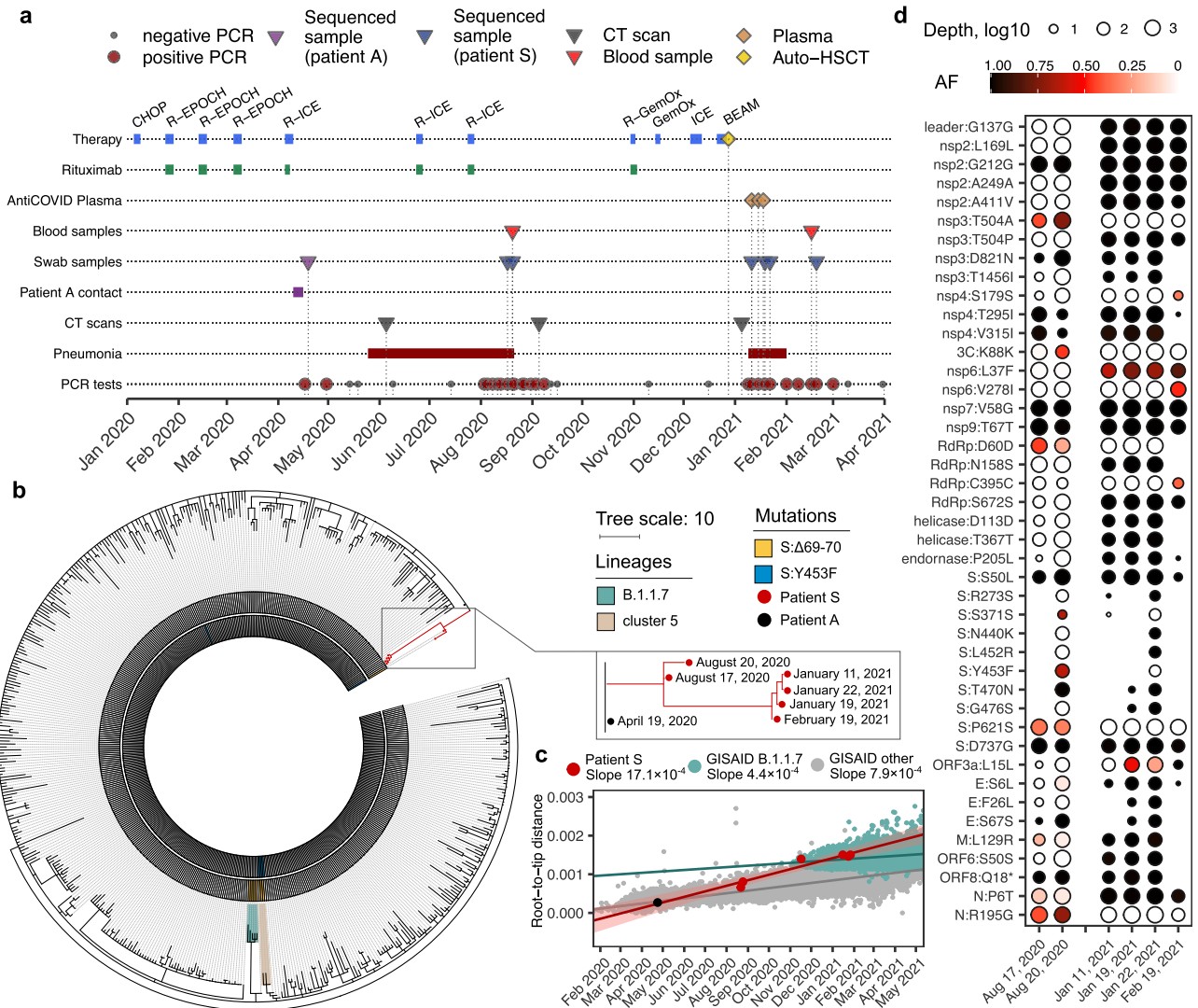

**Fig. 1 | Intrahost evolution of SARS-CoV-2 in patient S. a** The timeline of patient S disease and therapy. **b** The phylogenetic tree of B.1.1 pruned to contain a random set of 1% of all samples, including the patient A sample (black dot) and the complete clade carrying the patient S samples (red dots). The 2020 samples carried the ΔF combination of mutations (S:Δ69-70HV and S:Y435F; Supplementary Note 3) marked in the two inner circles in yellow and blue. The B.1.1.7 lineage and cluster 5 are shaded. **c** Regression of root-to-tip genetic distances vs. sampling dates, for patient S samples (together with the ancestral sample of Patient A), B.1.1.7 lineage

GISAID samples, and other GISAID samples. Shaded area around the regression curves represent 95% confidence interval. Estimated slopes (molecular clock rates) are provided in the inset. In **b** and **c**, the consensus nucleotides (i.e., those supported by more than 50% of the reads, RF > 50%) were used to position patient S and A samples. **d** Variant frequencies in the six patient S samples. All consensus variants (RF > 50%, N = 40) and nonconsensus variants with 30% < RF < 50% (N = 7) are shown (Supplementary Table 2). Source data are provided as a Source Data file.

autologous haematopoietic stem cell transplantation (auto-HSCT) was performed. In January 2021, near the end of the study period, patient S received three doses of convalescent plasma. Six nasopharyngeal swab samples suitable for next generation sequencing, together spanning 308 days of the disease, were obtained, alongside two blood samples (Supplementary Table 1).

## Intrahost evolution of SARS-CoV-2

Whole-genome sequencing was performed for six nasopharyngeal swab samples obtained from patient S in August 2020–February 2021, as well as for an April 2020 sample obtained from patient A (Fig. 1a). Phylogenetic analysis (Supplementary Note 2) indicates that both PCR positive periods of patient S in August 2020 and January-February 2021 constitute a single infection. Indeed, all patient S samples formed a single clade within the B.1.1 lineage on the global SARS-CoV-2 phylogeny, with the patient A sample as its ancestor (Fig. 1b). No other Russian samples available in GISAID nest within the patient S clade (Fig. 1b), indicating that the virus evolved in patient S has not seeded observable onward transmission.

The two August 2020 samples were characterized respectively by 12 and 18 mutations specific to patient S. In turn, the January-February 2021 samples gained additional 10 to 21 changes. Overall, a total of 40 changes compared to the ancestral state were observed in at least one of the samples, 34 of which were observed by the end of the study period (Supplementary Note 3). This corresponds to the point substitution rate of $15.3 \times 10^{-4}$ per nucleotide per year, which substantially exceeds the evolutionary rate of SARS-CoV-2 in the general population (permutation test, $p < 10^{-4}$; Fig. 1c). Nearly all accumulated changes were detected in samples obtained before convalescent plasma transfusions (Fig. 1a,d; Supplementary Table 1), indicating that these transfusions could not have affected the observed viral evolution.

The accumulated mutations occurred throughout the viral genome, affecting 18 of the 26 viral genes (Fig. 1d). However, there was an excess of nonsynonymous changes in the genes encoding surface proteins: out of the 25 changes, 8 (32%) fell in the spike gene which by length constitutes 13% of the viral genome, while 2 (8%) fell in the envelope gene which constitutes 0.8% of the genome (two-sided Binomial test, $p = 0.018$ and 0.016, respectively). Many of the observed amino acid substitutions were indicative of positive selection in the general population (Supplementary Note 4), and some were previously implicated in antibody escape (Supplementary Note 4). However, virus evolution did not lead to a detectable reduction in sensitivity to neutralizing antibodies by the end of the study period compared to a prototype viral strain (Supplementary Fig. 4).

## Host immune response

To understand the functional features of immune response in patient S, we analyzed her blood samples collected at multiple timepoints spanning the course of the disease (see Methods, Supplementary Table 1). Flow cytometry revealed the absence of B lymphocytes throughout the period of PCR positivity (Supplementary Fig. 5). Blood serum samples were also analyzed by ELISA for IgG antibodies specific to the SARS-CoV-2 S-antigen; a weak IgG response was registered in one of the samples but no response in the remaining samples. No neutralizing antibodies were detected at any time point by a VN assay using live SARS-CoV-2 strain (Supplementary Table 1).

By contrast, we detected a pronounced SARS-CoV-2 specific T-cell response. Indeed, in vitro stimulation with a peptide mixture of SARS-CoV-2 proteins (S, N, M, ORF3a, and ORF7a) caused an expansion of SARS-CoV-2-specific CD4 and CD8 effector memory T-cells (Tem) at both time points (Supplementary Fig. 6).

## Mutational escape from cytotoxic T cells

Given the absence of B-cell but the presence of T-cell immune response in patient S, we hypothesized that the 31 amino acid

sequence-altering mutations acquired by SARS-CoV-2 may have led to escape from T cell immunity. First, we asked if these mutations affect the presentation of the peptides carrying them by the HLA alleles of patient S (Supplementary Table 3). For this, we adapted an existing pipeline[30] to calculate the PHBR (patient harmonic best rank) score (Fig. 2a) for both the ancestral and the derived state at site of each of the 30 mutations (except ORF8:Q18*, Supplementary Note). Most sites could be presented in their ancestral state by at least one HLA allele of both classes (27 out of 30 by HLA I, and 24 out of 30 by HLA II). We found that five of the observed mutations substantially (>3-fold) increased the PHBR score for the peptides presented by HLA I, indicating impaired presentation (Fig. 2b). One of these mutations, S:del141-144, also increased the PHBR score for HLA II (Fig. 2c).

While an increase in PHBR score can help a peptide escape antigen presentation, this can only affect T cell response if the corresponding peptide is recognized by T cells. To specifically address the effect of mutations on immunogenic peptides, we used IEDB[31] to obtain the list of SARS-CoV-2 peptides that were shown to be immunogenic in complexes with the HLA alleles carried by patient S. There were 17 such peptides for HLA I alleles, together overlapping the sites of 11 of the mutations (some of the sites were covered by more than one peptide) (Supplementary Table 4). All these mutations were fixed in the course of intra-host evolution by the end of the study period. No HLA class II immunogenic peptides covering the changed sites were found in IEDB. To focus on the immunogenic peptides, we calculated the imBR (immunogenic best rank) for each of these sites in the ancestral state and compared it to the corresponding value for the derived state. The mutations strongly decreased presentation of immunogenic peptides, indicating that they cause escape from CD8 T cell response (Fig. 2d). Together with ORF8:Q18* which prevented presentation of the bulk of the ORF8 protein (Supplementary Note 5), this totals to 12 changes with cytotoxic T cell escape effect.

## Tracking the viral escape

Next, we assessed the change in T-cell response caused by the observed mutations. First, we focused on the two mutations causing the largest PHBR fold change (Fig. 2b). These were the two mutations at position 504 of the nsp3 protein, nsp3:T504A and nsp3:T504P, which were fixed sequentially at the first (T1, August 20, 2020) and the second (T2, February 19, 2021) sampled time points respectively (Fig. 1d). We asked how well the peptides covering these three amino acid variants elicited T-cell response in samples corresponding to these time points. We used the highest ranking peptides covering the mutated site in its ancestral (PTDNYITTY) and derived (PADNYITTY, PPDNYITTY) states; PTDNYITTY was previously shown to be immunogenic in complex with the HLA-A:01*01 allele which is carried by patient S[32–34].

In the T1 sample, when just nsp3:T504A was detected at intermediate frequencies (Fig. 1d), in vitro stimulation of CD8+ T cells indicated response to both the ancestral (PTDNYITTY) and the derived (PADNYITTY) peptide changed by nsp3:T504A (Fig. 3). This response was mediated primarily by polyfunctional IFNγ⁺IL2⁻TNFα⁺ effector memory T-cells. The response to PADNYITTY was slightly weaker (0.035% vs 0.043% of effector CD8 T cells), suggesting a partial escape caused by nsp3:T504A. Stimulation by PPDNYITTY corresponding to the nsp3:T504P allele caused no cytokine response in the T1 sample. In the T2 sample (Fig. 1a), when nsp3:T504P was already fixed, still no cytokine response to PPDNYITTY was observed, confirming invisibility of this peptide to cellular immune response due to weak binding with HLA. Response to PTDNYITTY and PADNYITTY also vanished at T2; this could indicate that the CD8 T cell clones specific to T and A amino acids became irrelevant with the loss of the corresponding viral variants, and got no antigenic re-stimulation that could drive clonal expansion after auto-HSCT[35].

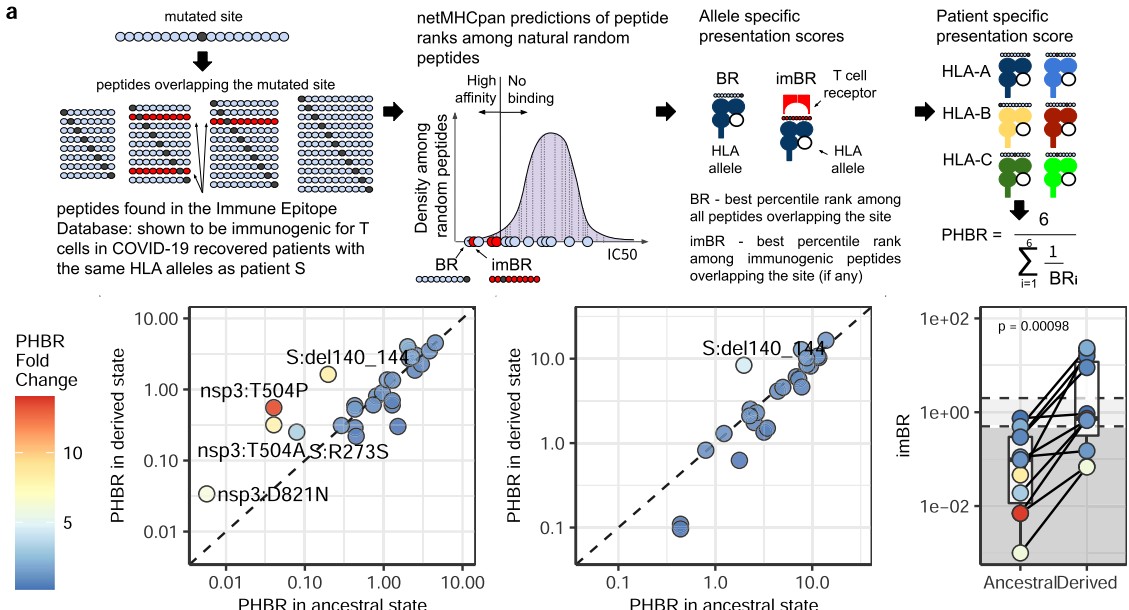

**Fig. 2 | Mutational escape from cytotoxic T cells. a** Calculation of site presentation scores (adapted from Marty et al.[30]). **b**, **c** Change of PHBR scores caused by mutations for HLA I **b** and HLA II **c**, respectively. Dot color corresponds to PHBR fold change; the mutations that substantially (>3-fold) increase PHBR are signed. Sites that did not bind any of the patient's HLA alleles both in ancestral and derived states are not shown. **d** Comparison of imBR scores for the mutated sites in their ancestral and derived states ($n = 11$). The level of significance is calculated by the two-sided paired Wilcoxon exact sign-rank test: Test statistic V = 0, p-value = 0.0009766, 95% confidence interval: [−11.7835; −0.3065], median estimates −4.647. Horizontal lines in a boxplot represent minima, first quartile, median, third quartile, and maxima. Source data are provided as a Source Data file.

Next, we explored the T-cell response to the pool of peptides corresponding to virus epitopes that gained amino acid mutations between August 2020 and January 2021. The pool included 5 peptides with previously confirmed immunogenicity and characterized by a strong change in PHBR due to the observed mutations, indicating a probable escape from the HLA alleles of patient S. We compared the pool of peptides in their ancestral state: YLQP**R**TFLL (S:R273S), S**T**NVTIATY (nsp3:T1456I), K**P**RSQMEIDF (endornase:P205L), G**P**QNQRNAPRITF (N:P6T) and VPLHGTI**L** (M:L129R), to the corresponding peptides with acquired amino acid mutations that resulted in weak or no binding: YLQP**S**TFLL, S**I**NVTIATY, K**L**RSQMEIDF, G**T**QNQRNAPRITF and VPLHGTI**R** (Supplementary Table 4).

At time point T1, we found a pronounced subpopulation of polyfunctional (IFNγ+/TNFα+) cytokine producing CD8 T cells responding to initial non-mutated peptides. This subpopulation comprised 0.95% of effector CD8 T cells, indicating a strong T cell response to this set of epitopes (Fig. 3c). Meanwhile, no T cell response was observed against the pool with acquired mutations, confirming immunoediting-driven origin of the observed mutations. Both peptide pools showed negligible response at time point T2, presumably due to poor post-HSCT expansion of T cell clones in the absence of the cognate antigenic stimulus. Prior to the escape, the CD8 T-cells responding to the pool of the 5 escaping peptides (0.95%), together with the peptide changed by nsp3:T504P (0.045%), constituted as much as ~1% of the total effector CD8 subset, and this response has been fully eliminated by viral escape.

**Possible population-level effects**

It has been suggested that escape from humoral immunity in immunosuppressed patients may give rise to SARS-CoV-2 strains with increased fitness in the general population[1]. Similarly, escape from cellular immunity in the course of intra-host evolution could affect immune response to descendant SARS-CoV-2 strains outside the host where it evolved. We aimed to estimate the possible effect of the viral evolution in patient S for the human population at large. For this, we compared the BR (Fig. 2a) fold change caused by the mutations

observed in patient S for the globally most frequent HLA alleles of each family that together cover 95% of worldwide population frequency[36,37]. This set of alleles includes all 12 HLA alleles of both classes (I and II) of patient S, which happen to be quite frequent globally (Supplementary Table 3).

As expected, the mutations observed in immunogenic epitopes tended to escape the HLA I alleles of patient S to a larger extent than other frequent HLA I alleles (Fig. 4a, b); no such difference was observed for HLA II alleles (Fig. 4c, d). Nevertheless, these same mutations also reduced binding for other globally frequent HLA I alleles (mean BR fold change = 1.59, Fig. 4e), although not HLA II alleles (mean fold change = 1.02, Fig. 4f). This indicates that the within-host evolution in patient S indeed could facilitate escape from cytotoxic T cells in the global population.

## Discussion

We have described a case of unprecedentedly long COVID-19 characterized by a large amount of intrahost evolution. For over 10 months, an evolving SARS-CoV-2 lineage accumulated changes at a rate which substantially exceeded that in the general population, suggesting prevalent viral adaptation. Some of the observed changes recapitulated mutations previously observed in other immunocompromised patients and/or variants of concern (Supplementary Fig. 7, Supplementary Note 3). This is consistent with the hypothesis that immunocompromised patients represent a hotspot of viral adaptation, causing "saltations" in the otherwise clock-like evolutionary rate of SARS-CoV-2;[1] notably, such a jump could have happened at the origin of the B.1.1.7 ("alpha") variant which has attained global dominance in early 2021[1,38].

Unlike previously described cases, however, the case described here is characterized by an unusual immune environment. The absence of own B cells, convalescent plasma therapy or monoclonal antibodies therapy during most of the study period indicates that the bulk of viral mutations have accumulated in the absence of humoral immune response. Instead, our data shows that evolution was largely driven by T cell escape. Our computational analysis revealed that many mutations changed the amino acid composition of known immunogenic

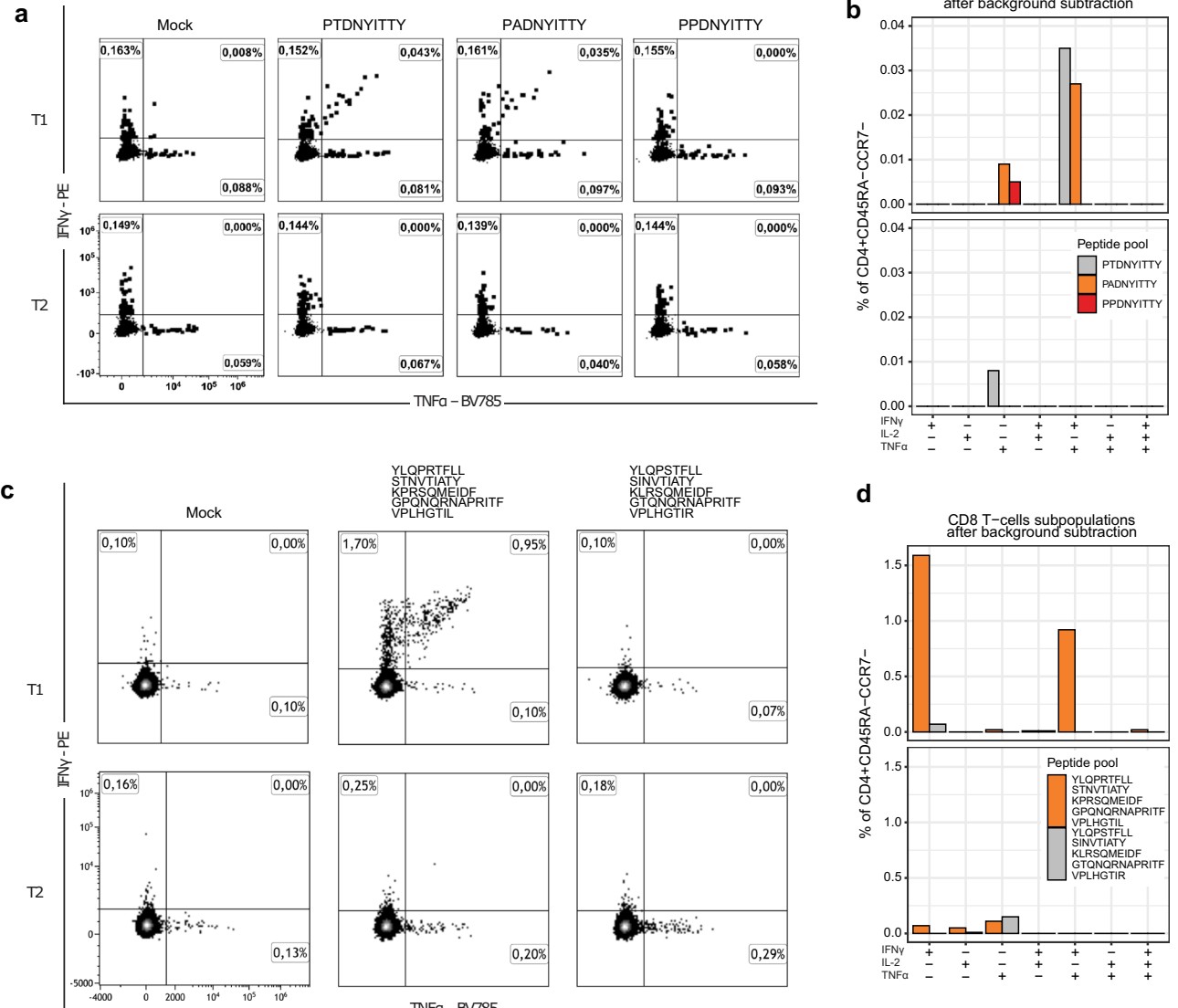

**Fig. 3 | The CD8 T cell immune response to the SARS-CoV-2 epitopes in ancestral and derived states. a, c** Flow cytometry plots showing the cytokine profiles of SARS-CoV-2-specific CD8 effector memory T cells after stimulation with epitopes carrying the ancestral and the two derived (nsp3:T504A and nsp3:T504P) amino acid variants **c**, and pools of 5 immunogenic HLA binders before and after acquiring the binder-escape mutations. Amino acid variants corresponding to ancestral and derived states highlighted by gray and red colors respectively. **b, d** Corresponding bar plots representing the percentage of different cytokine-producing populations of SARS-CoV-2-specific CD8 T cells after mock background subtraction. T1, August 2020 sample; T2, February 2021 sample. Source data are provided as a Source Data file.

CD8 T cell antigens and worsened or prevented their presentations on HLA class I alleles of the patient.

We experimentally tracked the viral escape by the 2 sequential mutations affecting the same amino acid position nsp3:T504 (nsp3:T504A and nsp3:T504P) and by the binder-escaping mutations in a pool of 5 immunogenic HLA binders (S:R273S, nsp3:T1456I, endornase:P205L, N:P6T and M:L129R). The elicited response in these two cases comprised 0.045% and 0.95% of all effector CD8 T cells, and this response has been eliminated by the escape. In a study of a cohort of 254 patients, the proportion of SARS-CoV-2 specific CD8 T cells in the overall IFN-γ expressing CD8 T cell pool rarely exceeded 1% and had the median of 0.2%[39,40]. Thus our experiments clearly demonstrate that the proportion of the CD8 + T cell response subject to viral escape is substantial.

Our study has certain limitations. Most importantly, it uses data from a single patient. Several features of our case make similar cases rare. First, long-term viral persistence in COVID-19 is relatively rare overall. Second, poor disease outcome is common in immuno-compromised patients[2–13], and such a long period of viral persistence and high amount of intra-host evolution is extreme even among ana-logous studies. Third, our case stands out among the others by the absence of convalescent plasma treatment during most of the disease. Nevertheless, in a recent preprint by Khatamzas and colleagues[41], apparent T cell escape was observed in a patient with follicular lym-phoma characterized by a shorter period of viral persistence and the usage of convalescent plasma. We consider that study an independent confirmation of our observations.

Another limitation is that our analysis only considered the effect of a subset of mutations. Our experimental measurements considered seven mutations, and disregarded other viral mutations that could have affected antigen presentation. Besides, some of the mutations may preserve antigen presentation but still alter the amino acid sequence of known CD8 T cell epitopes. Change of the epitope and prevention of recognition of the HLA-epitope complex by T cell

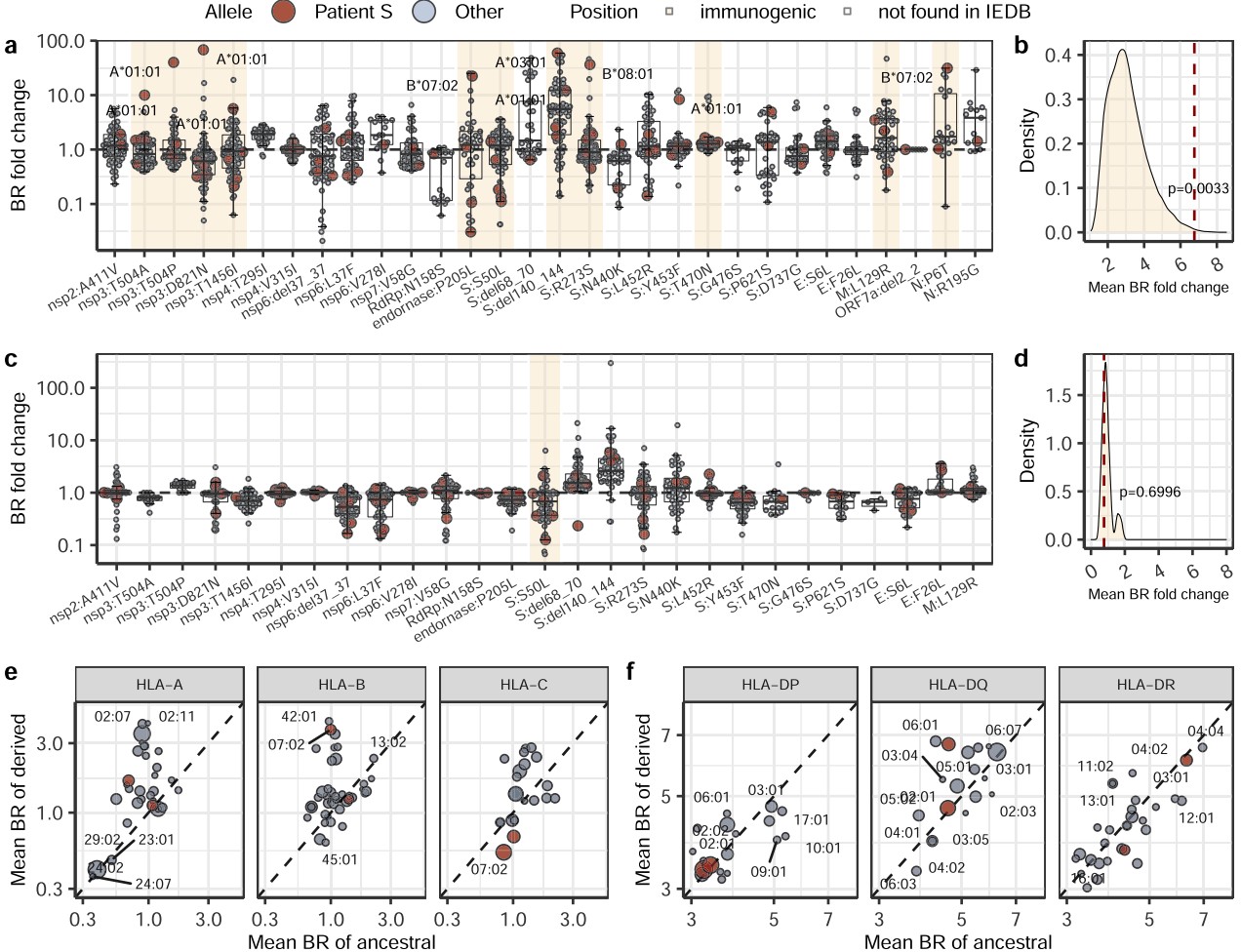

**Fig. 4 | Population-level effect of T cell escape mutations. a, c** The effect of each of the 30 mutations observed in SARS-CoV-2 of patient S on T cell immune escape, for each of the HLA I (**a**, n = 94) or HLA II (**c**, n = 71) alleles carried by patient A (red) and frequent globally (gray). The mutations that change immunogenic peptides (for HLA I) or are adjacent to such peptides (for HLA II) according to IEDB are highlighted. Alleles that do not present the corresponding position in both ancestral and derived state are not shown. For the mutations that correspond to >5-fold increase in BR, the corresponding HLA alleles are signed. Horizontal lines in a boxplot represent minima, first quartile, median, third quartile and maxima.

**b, d** Distribution of mean BPR fold changes among immunogenic positions for HLA I **b** or II **d** alleles, based on $10^5$ random generations of individual allele composition; the red dashed line is the percentile corresponding to the allele composition of patient S. One-sided level of significance is the following: p-value = 0.0033 for **b** and p-value = 0.6996 for **d**; **e, f** The sum effect of the amino acid changing mutations observed in SARS-CoV-2 of patient S on antigen presentation by the globally most frequent HLA class I **e** and class II **f**. The alleles of patient S are in red. Source data are provided as a Source Data file.

receptors was previously described as a mechanism of immune escape[25,26] and can also result in immune escape in our study. If additional escape is provided by these mutations, our appraisal of the proportion of the CD8 + T cell response negated by the viral escape is an underestimate.

Nevertheless, our results clearly indicate that immunoediting by cytotoxic CD8 clones is a prominent underappreciated factor in intrahost evolution of SARS-CoV-2. Similar to antibody escape, the T cell escape mutations acquired within an individual host may give rise to new epidemiologically important variants if they spill over to the general population. Notably, a recent study has revealed that CD8 T cell count is strongly associated with the level of intrahost diversity of the viral population in immunocompromised patients[13]. We predict that the changes observed in our study would also substantially affect SARS-CoV-2 antigenicity in the general population in case of onward transmission of the evolved variant. While no such transmission was detected in this case, our results emphasize an additional dimension of SARS-CoV-2 evolution which merits careful surveillance.

## Methods

### Ethics declaration

Our research complies with the Local Ethics Review Board of the Smorodintsev Research Institute of Influenza (approved on April 30, 2020, reference number: 131) and by the Biomedical Ethics Committee of the I.P. Pavlov First Saint Petersburg State Medical University. The research protocol was approved by IRB and ethics committees and participants gave written informed consent, according to CARE guidelines (See: https://www.care-statement.org/), and in compliance with the Declaration of Helsinki principles.

### Sample collection and sequencing

RT-PCR of swabs and sequencing of viral RNA were performed at the Smorodintsev Influenza Research Institute. All specimens were obtained and transported according to standard sampling protocol. RNA from nasopharyngeal swabs was extracted using QIAamp Viral RNA Mini Kit (QIAGEN). RNA from patient A post-mortem FFPE specimens was extracted using RNeasy FFPE Kit (QIAGEN). Samples were tested for SARS-CoV-2 viral RNA by real-time RT-PCR on thermal cycler

CFX96 (BioRad) using "Intifica SARS-CoV-2" Kit (Alkor Bio). Whole-genome amplification of SARS-CoV-2 virus genome for samples from August 2020 and from January 2021 was performed using BioMaster RT-PCR Premium kit (Biolabmix) and primers from ARTIC Network protocol version 3[42] and ARTIC Network protocol version 1[43] with modifications, respectively. Nextera XT (Illumina) kit was used for library preparation in August 2020 and DNA Prep (Illumina) kit was used for library preparation in January 2021, and the libraries were sequenced using the MiSeq platform (Illumina) with version 3 600-cycle chemistry.

The DNA of patient S was extracted from peripheral blood using QIAmp Blood DNA Mini kit. DNA sample was prepared and captured with the SureSelect Human All Exon kit v7 (Agilent), and whole exome was sequenced using MGISEQ-2000 at Pirogov Russian National Research Medical University (Moscow, Russia).

### Flow cytometry assays

Flow cytometry assays were performed using cryopreserved PBMCs. Cells were isolated from patients' heparinized blood by gradient centrifugation with lymphocyte separation medium Lymphosep (Bio-West), frozen in freezing medium containing 10% DMSO (AppliChem) in FBS (Gibco) and stored in liquid nitrogen until usage.

For B-cells analysis presented in Supplementary Fig. 5, PBMCs samples were towed in a 37°C water bath and stained with fluorescently-labeled antibodies to surface markers CD19-APC/Fire 750 (Clone: SJ25C1, Biolegend), BV421-CD20 (Clone: 2H7, Biolegend), CD3-BV605 (Clone: OKT3, Biolegend) (RRIDs and catalog numbers are provided in Supplementary Data, Reagents). PBMCs from a healthy volunteer were used as a control. B-cells were identified as a live CD3-/CD19 + /CD20 + population.

The T-cell response was assessed by intracellular cytokine staining. For further analysis, cells were towed in a 37°C water bath and stimulated for 6 hours with 5 μg/ml of the commercial available peptide mixture of SARS-CoV-2 proteins S, N, M, ORF3a and ORF7a (Generium, Russia)(for Supplementary Fig. 6b, c) or one of the peptides PTDNYITTY, PADNYITTY or PPDNYITTY or peptide pools (YLQPRTFLL + STNVTIATY + KPRSQMEIDF + GPQNQRNAPRITF + VPL-HGTIL and YLQPSTFLL + SINVTIATY + KLRSQMEIDF + GTQNQRNA-PRITF + VPLHGTIR) (for Fig. 3) in the RPMI medium (Gibco), containing 10% of FBS (Gibco), 1% of penicillin-streptomycin solution (Gibco), Brefeldin A (BD) and costimulatory CD28/CD49d reagent (BD). Negative control samples were stimulated with the complete medium; for positive control, PMA/ionomycin (Sigma) combination was used. Surface markers were stained with fluorescent antibody panel containing CD3-APC/Fire (Clone: SK7, Biolegend), CD4-AF647 (Clone: SK3, Biolegend), CD8a-AF700 (Clone: HIT8a, Biolegend), CD45RA-PE/Dazzle (Clone: HI100, Biolegend), CD197-BV421 (Clone: 150503, BD). Intracellular cytokines were stained using IL-2-FITC (Clone: MQ1-17H12, Biolegend), IFNγ-PE (Clone: 45.15, Beckman Coulter), TNFα-BV785 (Clone: MAb11, Biolegend) antibodies (RRIDs and catalog numbers are provided in Supplementary Data 1, Reagents). Cells were permeabilized with BD Cytofix/Cytoperm™ Fixation/Permeabilization Solution Kit (BD) according to the manufacturer's instructions. Data was collected on a CytoFlex flow cytometer (Beckman Coulter) using CytExpert software (Beckman Coulter). The results were analyzed using the Kaluza Analysis v2.1 program (Beckman Coulter). Interleukin (IL) 2, interferon γ (IFNγ) and tumor necrosis factor (TNFα) response was measured in effector memory T cells (Tem). To identify Tem, lymphocytes were gated based on their size and granularity. Live CD3⁺T cells were identified and subdivided into CD4 + and CD8 + T cells. These populations were further subdivided based on the expression of CD45RA and CD197(CCR7). CD3 + CD4 + or CD3 + CD8 + lymphocytes with the CD45RA-/CCR7- phenotype were considered Tem cells (Supplementary Fig. 6a). Cut-off values for the definition of cytokine-producing T cell responses stimulated with

SARS-CoV-2 peptides were ≥5 events and a ≥ 2-fold difference in the magnitude of TNF + , IFNγ + or IL-2+ Tem cells compared to the non stimulated control.

### Virus isolation and antigenicity

Live viruses (samples 30579 V and 30769 V from August 20, 2020 and 22748 V and 23680 V from February 19, 2020) were isolated from patient S swab samples in Vero E6 cells (IZSLER #BSCL87). Culture was inoculated for 2 hours with swab material diluted 1/10 in DMEM (Biolot) supplemented with 2% HI-FBS (Gibco), 1% anti-anti (Gibco) and then incubated for 3 days until first CPE signs. Samples were subsequently passaged one time in Vero cells (ATCC #CCL81). Virus culture media contained AlphaMEM (Biolot) supplemented with 2% HI-FBS (Gibco), 1% anti-anti (Gibco). (RRIDs and catalog numbers are provided in Supplementary Data 1, Reagents).

A total of 16 serum samples were obtained during the first wave of the COVID-19 pandemic in spring-summer 2020 from recovered volunteers with PCR-confirmed SARS-CoV-2 infection and tested in a microneutralization assay.

Microneutralization was performed with hCoV-19/St_Petersburg-3524V/2020 virus (GISAID EPI_ISL_415710, with the ΔF combination of mutations absent, designated as Reference), and 30769 V and 23680 V viruses isolated from the patient S (designated patient S August 2020 and patient S January 2021, respectively). Serum was heat inactivated (56 °C, 60 min), serially diluted 2-fold starting from 1/10, mixed with 25 TCID50 of virus, incubated for 1 h at 37 °C and inoculated into Vero cells in triplicates in 96-well plate. Culture media was the same as for virus isolation. 5 days after inoculation, neutralizing antibody titer was calculated as the reciprocal of the highest serum dilution preventing CPE.

Serum samples obtained from patient S were tested for virus specific antibodies in ELISA and in microneutralization assay with either Reference or patient S viruses. ELISA was performed with "SARS-CoV-2-IgG-EIA-BEST" (VEKTOR BEST #D-5501) according to the manufacturer's instructions.

### HLA genotyping

HLA genotyping was performed using a commercial kit according to the manufacturer's instructions (PARallele™ HLA solution v3, Parseq Lab). HLA-A, -B, -C, -DRB1 and -DQB1 loci were genotyped with 3-field resolution. Simultaneously, HLA calling was performed from WES data using HLA-HD version 1.3.0[44] with IPD-IMGT/HLA database Release Version 3.43. The inferred alleles are listed in Supplementary Table 3.

Using HLA-2-Haplo software tool[45] this set of alleles was split into two haplotypes presented in Supplementary Table 3. A European population database was used in this procedure. An a-posteriori probability of found combination was 97.6%. As one can see, the found haplotypes are among the most common variants in the European population.

### Consensus calling

Raw reads were trimmed with Trimmomatic version 0.39[46] to remove adapter sequences and low-quality ends. Trimmed reads were mapped onto the Wuhan-Hu-1 (MN908947.3) reference genome with BWA MEM version 0.7.17[47]. The following reads were then removed from the mapping: reads with abnormal insert length to read ratio (greater than 10 or less than 0.8), reads with insert length greater than 1100, reads with more than 50% soft-clipped bases. Soft-clipped ends were trimmed from the remaining reads, 10 nucleotides were cropped from read ends using custom scripts, and primer sequences were removed with ivar version 1.3[48]. Only reads with at least 30 nucleotides remaining after the procedure were kept. SNV and short indel calling was done with LoFreq version 2.1.5[49], with SNVs considered consensus if they were covered by at least 4 reads and supported by more than 50% of those reads; indels were considered consensus if they were covered by at least 20 reads with at least 50% of those supporting the variant.

Regions that were covered by fewer than 4 reads were masked as NC. We attributed several positions that were covered by 2 or 3 reads, but matched the reference and were conserved throughout all samples (22612, 23680, 24160, 27064, 30579 and 30769), to the reference; as these positions did not mutate, this decision did not affect any of our analyses. Consensus was created by bcftools version 1.9[50,51] consensus.

## Phylogenetic analysis

255,389 genomes of SARS-CoV-2 were downloaded from GISAID on December 12, 2020, (Supplementary Data) and aligned with MAFFT v7.453[52] against the reference genome Wuhan-Hu-1/2019 (NCBI ID: MN908947.3[53] with --addfragments --keeplength options. 100 nucleotides from the beginning and from the end of the alignment were trimmed. After that, we excluded sequences (1) with more than 300 positions of missing data (Ns) and gaps, (2) excluded by Nextstrain (https://github.com/nextstrain/ncov/blob/master/defaults/exclude. txt), or (3) from non-human animals other than minks, leaving us with 201,948 sequences. Identical sequences were then collapsed within the country and host and annotated by the Pangolin package version 2.1.0[54]. To this dataset, we added the two patient S samples obtained in August, 2020 as well as the patient A sample. As sequences of patient S belonged to the B.1.1 lineage, we further only kept sequences annotated as B.1.1, excluding a large clade defined by mutation G25563T (GH clade in GISAID[55] nomenclature). For the purposes of phylogenetic analysis, we additionally masked the highly homoplastic site 11083. The final set of 49,083 sequences was used to construct the phylogenetic tree with IQ-Tree v2.1.1[56] under the GTR substitution model and '-fast' option. Ancestral sequences at the internal tree nodes were reconstructed with TreeTime v. 0.8.0[57]. Having ensured that the two patient S samples form a clade rooted at the patient A sample and not carrying any samples other than those of patient S, we then separately reconstructed the phylogeny of all six samples of patient S, rooted it with patient A, and manually added the resulting clade to the downsampled B.1.1 tree. For visualization purposes, the tree was downsampled to contain 1% of samples, including the patient A sample and the complete clade containing all patient S samples.

To estimate the molecular clock rate of the patient S lineage (Fig. 1c), we downloaded all sequences available in GISAID on May 31, 2021, filtered them as described above, and subsampled the filtered dataset to 50,000 samples preserving all Russian sequences. To this dataset, we added the six patient S samples and the ancestral patient A sample. We then aligned the obtained 50,007 sequences against the reference sequence and reconstructed the phylogeny with Fasttree version 2.1.11[58]. Finally, we computed root-to-tip distances and calculated the slope of the root-to-tip distance vs. sampling dates regression line for the three separate datasets: (1) patient S samples, (2) B.1.1.7 samples, and (3) the remaining samples from the subsampled GISAID dataset. To validate the difference between the estimated clock rates for patient S samples and samples belonging to dataset (3), we subsampled this dataset, picking six random samples collected on the same dates as the patient S samples, and computed the linear regression slope, in each of the 10,000 trials. (For dataset (2), this procedure was impossible because there were no B.1.1.7 samples in August 2020). None of the 10,000 samples resulted in the estimated clock rate above $15.3 \times 10^{-4}$, implying the $p$-value of <0.0001.

## Effect of viral mutations on antigen presentation

To study the effect of mutations in SARS-CoV-2 proteins on their antigen presentation, we adapted metrics from Marty et al.[30] (Fig. 2a). For each mutated site in both its ancestral and derived states, we inferred all possible peptides of certain lengths overlapping it, and calculated their percentile ranks (Rank_El) relative to a set of random natural peptides by netMHCpan version 4.1 and netMHCIIpan version 4.0[59] for HLA I and HLA II respectively. We used peptide lengths between 8 and 12 amino acids for HLA I alleles, and between 12 and 18

amino acids for HLA II. If the mutated site was not presented by any of the HLA alleles either in the ancestral or derived states, we excluded it from analysis. To exclude non-presenting peptides, we used the percentile rank <2% threshold for HLA I, and <10% threshold for HLA II, as recommended by the netMHCpan manual. For derived states of deletions, we extended the peptide in the C-direction as necessary to preserve its length. We paired the predicted A and B chains of HLA class II alleles as suggested in the tool allele list: HLA-DQA10101-DQB10501, HLA-DQA10501-DQB10201, HLA-DPA10103-DPB10402, HLA-DPA10103-DPB10401, DRB1_0301, DRB1_0101. We excluded the stop-codon producing mutation ORF8:Q18* from comparisons of ancestral and derived states, since the corresponding values for the derived state were undefined.

As in Marty et al.[30], we used the best percentile rank (BR) among all possible peptides overlapping the mutated site as the presentation score of this site for the particular HLA allele. To estimate the overall presentation of the site in the patient, we calculated the patient harmonic best rank (PHBR), i.e., the harmonic mean of BRs of HLA alleles of the same class. To compare the effect of a mutation on site presentation, we calculated the fold change of PHBR score as the ratio of the derived PHBR to the ancestral PHBR (so that fold change > 1 indicates weakening of presentation).

To focus on the peptides shown to be immunogenic to T cells in other SARS-CoV-2 infected patients carrying the same HLA alleles as patient S, we used IEDB[31] (Immune Epitope Database and Analysis Resource, accessed on June 1, 2021) with the "positive assay only" filter. IEDB identifiers of found epitopes are presented in the Supplementary Data file. For those sites inferred to be contained in immunogenic peptide, we calculated the best percentile rank of immunogenic peptide overlapping the site of mutation (imBR).

## Population-level effects of mutations

To check the effect of detected SARS-CoV-2 mutations on presentation by the HLA alleles other than those of patient S, we calculated the BR scores as explained above for the most frequent classical HLA alleles of each family that together represented 95% of the HLA alleles in the world population. The list and frequencies of such alleles were taken from Sarkisova et al. and Solberg et al.[36,37].

For most mutations detected in immunogenic epitopes, at least one of the HLA I alleles of patient S demonstrated extreme values of BR fold change in comparison with other alleles (Fig. 4a). To check the probability of such an observation happening by chance, we performed a permutation test, calculating the probability that a randomly chosen set of alleles has the same or a more extreme value of mean BR fold change across all mutations overlapped by immunogenic peptides as that of alleles of patient S. This was true for 33 out of 100000 permutations, corresponding to $p = 0.0033$ (Fig. 4b). None of the HLA II immunogenic epitopes overlapped any of the mutated sites; the only mutated site adjacent to such an epitope (S:S50L) did not stand out in the permutation test ($p = 0.6996$; Fig. 4c, d).

To compare the effects of mutations between different HLA alleles in Fig. 4e, f, we calculated the mean BR across all changed sites. This analysis again excluded ORF8:Q18*, which nevertheless prevented production of high-affinity epitopes for most alleles.

## Data analysis and visualization

Data analysis was performed in R version 4.0.0[60], and figures were visualized with ggplot2 package version 3.3.2[61]. SARS-CoV-2 phylogenetic tree was visualized with ITOL version 6[62].

## Statistics and reproducibility

No statistical method was used to predetermine the sample size. No data were excluded from the analyses. The experiments were not randomized. The Investigators were not blinded to allocation during experiments and outcome assessment.

## Reporting summary

Further information on research design is available in the Nature Research Reporting Summary linked to this article.

## Data availability

The SARS-CoV-2 raw sequencing data generated in this study have been deposited in the Sequence Read Archive under accession code SRA: PRJNA749008 (Supplementary Data). The processed consensus genome sequences of SARS-CoV-2 are available at GISAID's EpiCoV database under accession code EPI_SET_221010fc (DOI: 10.55876/gis8.221003dg). Accession identifiers of separate samples are available in the Supplementary Data file. Accession codes of other SARS-CoV-2 genome sequences, downloaded from GISAID and used in the study, as well as identifiers of T cell epitopes, downloaded from IEDB, are provided in the Supplementary Data file. Source data are provided in this paper.

## Code availability

Code is available at https://github.com/EvgeniiaAlekseeva/patient_S (https://doi.org/10.5281/zenodo.77149702).

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

## Acknowledgements

We are thankful to patient S for participation in this study. This study was supported by RFBR project 20-04-60556 to G.A.B. E.I.A. supported by RFBR, project number 20-34-90153. D.M.C. was supported by a grant from the Ministry of Science and Higher Education of the Russian Federation (075-15-2019-1789). D.M.D, A.B.K. and D.A.L. were supported by the Governmental contract "Genetic Analysis of SARS-CoV-2 in Russia in the first wave of COVID-19 pandemic". Part of this study performed at Smorodintsev Research Institute of Influenza was funded by the Russian Ministry of Science and Higher Education as part of the World-class Research Center program: Advanced Digital Technologies (contract No. 075-15-2022-313, dated 20.04.2022). The authors acknowledge the Resource Center "Center Biobank" (St. Petersburg State University) for technical support. We are thankful to Mikhail Shugay, Aleksandr Tashkeev, and Vadim Karnaukhov for their helpful discussions. We thank all of the authors who have contributed genome data on GISAID (see Supplementary Data for the list).

## Author contributions

O.V.S. provided the detailed clinical picture and participated in study design; E.I.A. designed and performed the analysis of mutation escape effects; M.S. cultured the virus and performed neutralization assay; A.V.F., K.S.K., and A.A.I. produced sequencing data; T.S.S. performed HLA genotyping; K.A.V., A.-P.S. and M.A.S. performed T-cells assays; K.R.S., E.R.N., S.K.G., G.V.K., and G.A.B. designed and performed genome analysis; K.R.S. analyzed the evolutionary rate; E.A.B. performed HLA-calling and HLA-haplotyping; J.V.Z. collected samples and communicated with the hospital; A.N.K. described the clinical picture and formalized the patient agreement; O.V.L. described the CT scans; I.A.S. is the patient's attending doctor; V.V.R. provided detailed diagnostic and treatment information about lymphoma; N.V.M. coordinated screening for SARS-CoV-2 between the oncohematology department and the reference laboratory; D.A.L. and G.A.B. provided coordination, supervision and funding acquisition; D.M.D. participated in study design and logistics; A.B.K., D.M.C., and G.A.B. planned the study; O.V.S., E.I.A., K.R.S., E.R.N., S.K.G., A.B.K., D.M.C., and G.A.B. drafted the manuscript; E.I.A. and G.A.B. wrote the manuscript, with contributions from all authors.

## Competing interests

The authors declare no competing interests.
