## [Peer Review File · Nature Communications]

REVIEWER COMMENTS

Reviewer #1 (Remarks to the Author):

Stanevich et al. report a case of prolonged SARS-CoV-2 infection in a patient with non-Hodgkin lymphoma receiving Rituximab treatment. The authors carefully analyzed the in-host viral evolution and relate the observed mutations to T cell immunity. This is an interesting study that adds to the body of work with respect to immune-driven SARS-CoV-2 evolution underpinning the potential of CD8+ T cells to mediate viral immune escape in addition to viral escape pushed by the antibody response. Most of the presented data rely on in silico prediction accomplished by in vitro T cell assays testing cytokine production in response to peptide stimulation. The manuscript is well written and clearly structured. However, there are some concerns that require attention prior to publication:

1.) The study comprises a single case. Thus, the overall relevance of T cell-driven viral escape in prolonged SARS-CoV-2 infection in general and in particular in Rituximab-treated patients remains elusive. This point should at least be discussed.

2.) The authors predict T cell epitopes corresponding to the HLA alleles of the patient covering the site of the established amino acid changes, however, this is no formal prove that these epitopes are indeed targeted in the patient. This should be stated accordingly. Formal proof is only given for one nsp3 epitope with the highest PHBR (thus also the highest likelihood for actual escape) and the sequential change of amino acids in this epitope. By focusing on this single epitope, it remains unclear which proportion of the CD8+ T cell response is subject to viral escape. Thus, what is the relevance of viral escape for the failure of the CD8+ T cell response to control the SARS-CoV-2 infection or vice versa what is the overall potential of the CD8+ T cell response to drive viral escape? Please add more detailed experimental data on the overall CD8+ T cell response including additional testing of viral escape.

3.) The background cytokine production of T cells in response to mock stimulation seems to be rather high. What is the reason for this? Please provide the complete gating strategy. Along this line, it is not clear to this reviewer how the authors define TEM cells in these experiments. Please indicate the pre-gating/used definition of TEM subsets more precisely. Furthermore, the authors apply short-term stimulations to test for T cell reactivity. However, this approach is not suitable to detect low-frequency CD8+ T cells. Peptide-specific in vitro expansion of T cells prior to testing of cytokine production after restimulation increases the detection sensitivity and therefore allows a more comprehensive assessment of the T cell reactivity.

Minor points:

1.) As patient S is continuously treated with cytotoxic regimens that possibly affect the overall immune status of the patient, a parameter like longitudinal leukocyte count would be helpful to rate the potential overall T cell pressure.

2.) Please provide additional patient data with relevance for the course of infection like patient blood group, BMI, etc.

3.) line 52 " cytotoxic T <cell> clones": "cell" is missing.

4.) Please check reference to the figures e.g., figure 4.

Reviewer #2 (Remarks to the Author):

The authors present a clearly described, thorough and carefully conducted case study of viral evolution and T-cell escape in a single patient. The patient received treatments including monoclonal therapy with rituximab and convalescent plasma, but remained SARS-CoV-2 positive for 308 days. The study brings together different data modalities, including genome sequence data for multiple samples from the patient as well as a sample from the likely infector, and contextualises the results within the broader phylogeographic structure of SARS-CoV-2 in the relevant timeframe and geographic region. An attempt is made to estimate the intrahost evolutionary rate; although data of necessity refer to a single patient, it is nevertheless a valuable addition to a growing body of literature in this field.

The authors compare sensitivity of the cultured virus to antibodies derived from convalescent plasma from 16 donors who experienced infection with a closely related circulating strain. Interestingly, the early isolate from the patient exhibits reduced sensitivity to antibodies, despite the near-absence of functional antibody response in the patient, while the late isolate does not. While this element of the work is less relevant given the patient's clinical presentation, it is nonetheless important in confirming that the subsequent novel findings are indeed related to the isolated T-cell response.

The authors further carry out flow cytometry assays to measure presence of distinct lymphocyte populations, and confirm that only T lymphocytes are detected (again, as might be expected from the clinical presentation). In a novel analysis, they carry out HLA genotyping, and show that the observed amino acid substitutions in the patient's virus are expected to affect antigen presentation by HLA I, suggesting that at least 12 of the observed viral mutations would be expected to result in T cell escape. Intriguingly, two of the mutations with the strongest effect identified were at the same site (nsp3:504), and were fixed sequentially at two different timepoints, consistent with ongoing selection pressure.

I have only minor comments, related to the genomics methods.

*consensus calling - it is unclear why positions with 2-3 depth were allowed to be called as Reference, but disallowed if Variant (unlike human genomics, the reference assumption is not strong and the same thresholds should be applied regardless of whether the call is variant or reference).

*Phylogenetic analysis - please clarify exclusion criteria: point (1) states that sequences shorter than 29,000 were dropped. After exclusion of leading and trailing 100bp, the total genome length should be 29703, so this allows 703 gaps/Ns. However, point (2) states that sequences with 300+ gaps/Ns were disallowed. Which threshold was applied? What does (3) "excluded by Nextstrain" mean here? (5) resequencing of same patients - this is impossible to identify reliably from GISAID data

*If highly homoplasic site 11083 was excluded, why was it included in the list of variants in Extended Data table 2?

*Fig1d - del37/variant L37F - presence of both as intrahost variants suggests variable mapping of this region in different reads so perhaps only include one of these in mutation counts, after checking the mapping.

*Extended data table 3 mentions "conventional genotyping" in the legend, but I could not find this in the methods.

Reviewer #3 (Remarks to the Author):

While directed evolution of SARS-CoV-2 has been described in promoting escape from neutralizing antibodies, evasion of cellular immunity via viral evolution has not, to my knowledge, been directly demonstrated. The authors describe preferential accumulation of nonsynonymous variants in regions of the genome encoding surface proteins, including spike and envelope genes. They subsequently utilized patient harmonic best rank scores to predict the presentation scores of variants of interest and determined that multiple mutations resulted in reduced predicted MHC-I presentation. They directly validated this by re-stimulating T-cells with variants of an nsp3-derived peptide showing complete loss of CD8+ T-cell responses to the T104P variant. Finally, they speculate that the variants that accumulate in this patient might not only promote immune escape in the host, but potentially escape within the global population based on analysis of the HLA alleles present in the majority of the population.

The manuscript is well written and supports a discrete hypothesis with excellent experimental design and analysis. It reports a finding of considerable biological and clinical interest and I am in favor of publication. Intriguingly, in a recent publication demonstrating prolonged COVID-19 infection in patients receiving B-cell depleting therapies, increased viral entropy was specifically noted in patients with impaired CD8+ T-cell immunity, and this manuscript would be worth citing (Lee et al, Cancer Discovery 2021).

Response: We thank the reviewers for their comments, which are addressed below.

REVIEWER COMMENTS

Reviewer #1 (Remarks to the Author):

Stanevich et al. report a case of prolonged SARS-CoV-2 infection in a patient with non-Hodgkin lymphoma receiving Rituximab treatment. The authors carefully analyzed the in-host viral evolution and relate the observed mutations to T cell immunity. This is an interesting study that adds to the body of work with respect to immune-driven SARS-CoV-2 evolution underpinning the potential of CD8+ T cells to mediate viral immune escape in addition to viral escape pushed by the antibody response. Most of the presented data rely on in silico prediction accomplished by in vitro T cell assays testing cytokine production in response to peptide stimulation. The manuscript is well written and clearly structured. However, there are some concerns that require attention prior to publication:

1.) *The study comprises a single case. Thus, the overall relevance of T cell-driven viral escape in prolonged SARS-CoV-2 infection in general and in particular in Rituximab-treated patients remains elusive. This point should at least be discussed.*

Response: We agree that expansion of the study to more patients would be informative. However, the specific features of therapy design here (rituximab + absence of convalescent plasma during most of the therapy) together with the longevity of the disease (this is one of the longest cases described to date) make such cases very rare and hard to observe. In revision, we learned about a new preprint (Khatamzas et al. 2022, line 278) also demonstrating a T cell escape during long-term COVID19. While the viral infection in this case was half as short as in

our patient and the T cell escape was detected using other methodology, we consider this work as an independent corroboration of our findings. Nevertheless, we agree that as a case study, it cannot be immediately extrapolated to other instances. We now add a section on limitations of our study to Discussion, and mention this point there (line 272-288).

2.) *The authors predict T cell epitopes corresponding to the HLA alleles of the patient covering the site of the established amino acid changes, however, there is no formal prove that these epitopes are indeed targeted in the patient. This should be stated accordingly. Formal proof is only given for one nsp3 epitope with the highest PHBR (thus also the highest likelihood for actual escape) and the sequential change of amino acids in this epitope. By focusing on this single epitope, it remains unclear which proportion of the CD8+ T cell response is subject to viral escape. Thus, what is the relevance of viral escape for the failure of the CD8+ T cell response to control the SARS-CoV-2 infection or vice versa what is the overall potential of the CD8+ T cell response to drive viral escape? Please add more detailed experimental data on the overall CD8+ T cell response including additional testing of viral escape.*

Response: To address this comment, we now obtained new experimental data. First, to more fully characterize the overall CD8 T cell response to cumulative stimulation, we used another commercially available peptide mixture (Generium, Russia) which covers more SARS-CoV-2 proteins (S, N, M, ORF3a and ORF7a) than the previously used mixture which only covered the N protein and the RBD region of the S protein (line 332, Extended Data Fig. 6b). Stimulation revealed response in both CD4 and CD8 T cell populations. The lower response of the CD8 compared to the CD4 T cells is possibly due to the extended length of peptides in the peptide pool: 15-mer sequences were used but 8-mer peptides are optimal for CD8 T-cell stimulation.

Second, we performed an additional experiment which provides further support for viral escape (lines 188 - 206, line 334, Fig. 3 c,d). As we were limited in the amount of the material from the patient, we were unable to study individual peptides further. Instead, we explored the T-cell response to the pool of peptides corresponding to viral epitopes that gained amino acid mutations between August and January. For this, we tested a pool of five peptides characterized by a high PHBR, indicating probable strong binding to Patient's HLA alleles, in their ancestral state: YLQPRTFLL (S:R273S), STNVTIATY (nsp3:T1456I), KPRSQMEIDF (endornase:P205L), GPQNQRNAPRITF (N:P6T protein) and VPLHGTIL (M:L129R protein) (Extended Data Table 4). We compared it with the pool that included the corresponding peptides with acquired amino acid changes that resulted in weak or no binding: YLQPSTFLL, SINVTIATY, KLRSQMEIDF, GTQNQRNAPRITF and VPLHGTIR. As expected, we found a pronounced subpopulation of polyfunctional (IFN γ +/TNF α +) cytokine-producing CD8 T-cells specific to the pool of peptides with the ancestral amino-acid variants (Fig. 3c). This subpopulation comprised 0.95% of effector CD8 T cells, indicating a strong T cell response to this set of epitopes. By contrast, the pool of peptides with acquired amino-acid changes induced no response, proving the escape effect of these changes. Both peptide pools showed a negligible response at time point T2, presumably due to poor post-HSCT expansion of T cell clones in the absence of the cognate antigenic stimulus.

It is challenging to measure the proportion of the CD8 T cell response subject to viral escape, mainly because we do not know the fraction of the CD8 T cells involved in the response to SARS-CoV-2 infection in our case. Still, some rough estimates can be made. Prior to the

escape, the CD8 T-cells responding to the pool of the 5 escaping peptides (0.95%), together with the peptide changed by nsp3:T504P (0.045%), constituted as much as ~1% of the total effector CD8 subset, and this response has been fully eliminated by viral escape. Previous work has shown that this value is comparable to the proportion of the entire CD8 T cells population involved in response to SARS-CoV-2, since the estimated median frequency of SARS-CoV-2 specific IFN- γ expressing CD8 T cell pool equals just 0.2% (Cohen et al. 2021). Although we did not measure the anti-viral CD8 T cell response to all possible epitopes comprehensively, this experiment clearly demonstrates that a very substantial proportion of the CD8 T cell response is subject to viral escape.

Cohen KW, Linderman SL, Moodie Z, *et al.* Longitudinal analysis shows durable and broad immune memory after SARS-CoV-2 infection with persisting antibody responses and memory B and T cells. *Cell Rep Med* 2021; 2: 100354.

3.) *The background cytokine production of T cells in response to mock stimulation seems to be rather high. What is the reason for this? Please provide the complete gating strategy. Along this line, it is not clear to this reviewer how the authors define TEM cells in these experiments. Please indicate the pre-gating/used definition of TEM subsets more precisely. Furthermore, the authors apply short-term stimulations to test for T cell reactivity. However, this approach is not suitable to detect low-frequent CD8+ T cells. Peptide-specific in vitro expansion of T cells prior to testing of cytokine production after restimulation increases the detection sensitivity and therefore allows a more comprehensive assessment of T cell reactivity.*

Response: The background cytokine production of T cells in response to mock stimulation is a variable characteristic and could take different values among individuals.

The Tem population was identified as follows: lymphocytes were gated based on their size and granularity. Live CD3+ T-cells were identified and subdivided into CD4+ and CD8+ T-cells. These populations were further subdivided based on the expression of CD45RA and CD197(CCR7). CD3+ CD4+ or CD3+ CD8+ lymphocytes with the CD45RA-/CCR7- phenotype were considered Tem cells. We clarified our procedure for Tem identification in the Methods section (line 352) and added a gating strategy in Extended Data Fig. 6a.

Methods involving enrichment of the T-cell population or expansion of a single T cell usually require a large amount of test material and are highly affected by multiple hardly predictable parameters such as the functional status of the cells, the presence of other reactive cells, the relationship between the number of cell divisions and cell death. Furthermore, the phenotype and function of the expanded cells may be significantly altered by culture conditions. Therefore, in this work, to determine the antigen-specific T-cell response, we have applied the most commonly used and robust method based on flow cytometry.

Minor points:

1.) *As patient S is continuously treated with cytotoxic regimens that possibly affect the overall immune status of the patient, a parameter like longitudinal leukocyte count would be helpful to rate the potential overall T cell pressure.*

Response: We added the dynamics of patient's lymphocytes, neutrophils and white blood cell counts in Extended Data Fig. 2 and mentioned it in the main text (lines 72 - 74).

2.) *Please provide additional patient data with relevance for the course of infection like patient blood group, BMI, etc.*

Response: Additional information about patient's BMI, blood group and concomitant diseases is added in the Supplementary Note 1, which is mentioned in the main text (line 58, lines 687 - 698).

3.) *line 52 " cytotoxic T <cell> clones": "cell" is missing.*

Response: added (line 48).

4.) *Please check reference to the figures e.g., figure 4.*

Response: checked, thank you for spotting this. However with the addition of new figures a few references have been changed (highlighted by yellow).

Reviewer #2 (Remarks to the Author):

The authors present a clearly described, thorough and carefully conducted case study of viral evolution and T-cell escape in a single patient. The patient received treatments including monoclonal therapy with rituximab and convalescent plasma, but remained SARS-CoV-2 positive for 308 days. The study brings together different data modalities, including genome sequence data for multiple samples from the patient as well as a sample from the likely infector, and contextualises the results within the broader phylogeographic structure of SARS-CoV-2 in the relevant timeframe and geographic region. An attempt is made to estimate the intrahost evolutionary rate; although data of necessity refer to a single patient, it is nevertheless a valuable addition to a growing body of literature in this field.

The authors compare sensitivity of the cultured virus to antibodies derived from convalescent plasma from 16 donors who experienced infection with a closely related circulating strain. Interestingly, the early isolate from the patient exhibits reduced sensitivity to antibodies, despite the near-absence of functional antibody response in the patient, while the late isolate does not. While this element of the work is less relevant given the patient's clinical presentation, it is nonetheless important in confirming that the subsequent novel findings are indeed related to the isolated T-cell response.

The authors further carry out flow cytometry assays to measure presence of distinct lymphocyte populations, and confirm that only T lymphocytes are detected (again, as might be expected from the clinical presentation). In a novel analysis, they carry out HLA genotyping, and show that the observed amino acid substitutions in the patient's virus are expected to affect antigen presentation by HLA I, suggesting that at least 12 of the observed viral mutations would be expected to result in T cell escape. Intriguingly, two of the mutations with the strongest effect identified were at the same site (nsp3:504), and were fixed sequentially at two different timepoints, consistent with ongoing selection pressure.

I have only minor comments, related to the genomics methods.

**consensus calling - it is unclear why positions with 2-3 depth were allowed to be called as Reference, but disallowed if Variant (unlike human genomics, the reference assumption is not strong and the same thresholds should be applied regardless of whether the call is variant or reference).*

Response:

In fact, all positions with coverage depth 2-3 (22612, 23680, 24160, 27064, 30579 and 30769) were conserved throughout all our SARS-CoV-2 samples (see Extended Data Table 2), matching the reference, and thus did not participate in the observed viral evolution. We technically attributed them as Reference; however, the way of their classification (Reference or NC) plays no role in our analysis or conclusions. We now clarified this in Methods, lines 402 - 405.

**Phylogenetic analysis - please clarify exclusion criteria: point (1) states that sequences shorter than 29,000 were dropped. After exclusion of leading and trailing 100bp, the total genome length should be 29703, so this allows 703 gaps/Ns. However, point (2) states that sequences with 300+ gaps/Ns were disallowed. Which threshold was applied? What does (3) "excluded by Nextstrain" mean here? (5) resequencing of same patients - this is impossible to identify reliably from GISAID data*

Response: Thank you for pointing this out; indeed, (2) covers (1). Points (3) and (5) refer to the list provided by Nextstrain (the link is now added). Resequenced samples are indeed impossible to identify reliably; we were referring to the duplicates identified by Nextstrain. The criteria are now rephrased more clearly (lines 411 - 414).

**If highly homoplastic site 11083 was excluded, why was it included in the list of variants in Extended Data table 2?*

Response: Site 11083 was excluded for the purpose of phylogenetic reconstruction; however, we had no reason to suspect the robustness of the two mutations called at this site in our sample, nsp6:L37F and nsp6:del37, so we did not deviate from our procedure of using them for analysis of variability. In Extended Data Table 2 and Fig. 1d, we show both consensus variants and variants with read frequency > 30%, to fully illustrate intrahost viral variability.

**Fig1d - del37/variant L37F - presence of both as intrahost variants suggests variable mapping of this region in different reads so perhaps only include one of these in mutation counts, after checking the mapping.*

Response: Only one of these variants (nsp6:L37F) reached 50% read frequency and was included in the consensus of the corresponding sample. Therefore, the Reviewer's suggestion is followed by our current procedure. In Fig. 1d, we show all variants reaching at least 30% read frequency; both these mutations match this criterion so both are shown. The exact read frequencies of each variant and their inclusion in the consensus sequence are listed in Extended Data Table 2. We added a link to it in the description of Fig. 1d (line 118).

**Extended data table 3 mentions "conventional genotyping" in the legend, but I could not find this in the methods.*

Response: We referred to HLA genotyping carried out with PARAllele™ HLA solution v3 kit (Parseq Lab). In the revision, we remove the word "conventional". Corresponding clarifications are made to the Table 3 description and to the HLA genotyping section of the Methods (line 845).

Reviewer #3 (Remarks to the Author):

While directed evolution of SARS-CoV-2 has been described in promoting escape from neutralizing antibodies, evasion of cellular immunity via viral evolution has not, to my knowledge, been directly

demonstrated. The authors describe preferential accumulation of nonsynonymous variants in regions of the genome encoding surface proteins, including spike and envelope genes. They subsequently utilized patient harmonic best rank scores to predict the presentation scores of variants of interest and determined that multiple mutations resulted in reduced predicted MHC-I presentation. They directly validated this by re-stimulating T-cells with variants of an nsp3-derived peptide showing complete loss of CD8+ T-cell responses to the T104P variant. Finally, they speculate that the variants that accumulate in this patient might not only promote immune escape in the host, but potentially escape within the global population based on analysis of the HLA alleles present in the majority of the population.

The manuscript is well written and supports a discrete hypothesis with excellent experimental design and analysis. It reports a finding of considerable biological and clinical interest and I am in favor of publication. Intriguingly, in a recent publication demonstrating prolonged COVID-19 infection in patients receiving B-cell depleting therapies, increased viral entropy was specifically noted in patients with impaired CD8+ T-cell immunity, and this manuscript would be worth citing (Lee et al, Cancer Discovery 2021).

Response: Thank you for pointing us to this very interesting study which is complementary to our results. We now mentioned it in the Introduction (line 39) and refer to it in Discussion (lines 293 - 294).

REVIEWERS' COMMENTS

Reviewer #1 (Remarks to the Author):

The authors satisfactorily addressed my concerns.